# PARTIAL-CORRELATION LEARNING FOR LARGE LANGUAGE MODELS WITH SKIP-TUNING

## ABSTRACT

Large Language Models (LLMs) require post-training to adapt to specific applications, with Supervised Fine-Tuning (SFT) crucial for injecting emerging or domain-specific knowledge. Conventional SFT using complete sequential text risks causing a distribution shift from pretraining corpora due to large volumes of common-style text, potentially leading to overfitting and catastrophic forgetting. We introduce Skip-Tuning, a novel fine-tuning strategy that utilizes non-successive text segments instead. Skip-Tuning performs skipped language modeling on text segments and enables a paradigm of partial-correlation learning, where the model learns from sparse but meaningful text fragments. By excluding common-style texts and using only knowledge-intensive text for fine-tuning, Skip-Tuning demonstrates improvements in fine-tuning effectiveness and generalization in the knowledge editing setting. Furthermore, we demonstrate the effectiveness of partial-correlation learning in a system-prompt following task, which illustrates the broad application of Skip-Tuning across various NLP scenarios.

## 1 INTRODUCTION

Large Language Models (LLMs) exhibit remarkable proficiency across a variety of natural language tasks. To tailor pretrained LLMs for specific real-world applications, post-training is an essential step. The prevailing view holds that LLMs primarily acquire knowledge during the pre-training phase. Post-training, which encompasses Supervised Fine-Tuning (SFT) for style and format adaptation (Ouyang et al., 2022; Zhou et al., 2023a) and Reinforcement Learning (RL) techniques for preference alignment (Rafailov et al., 2024; DeepSeek-AI, 2025), serves to unlock the model's inherent capabilities.

However, beyond style and format adaptation, SFT approaches are needed to inject knowledge that aligns with emerging or long-tail domain-specific knowledge(Huang et al., 2025; Han et al., 2024). Current practices in obtaining SFT data generally involve collecting or synthesizing "instruction-response" data that represents the target knowledge with complete sequential text. Furthermore, to accommodate downstream use cases, chat templates or other specific formats are generally employed (Illustrated in Figure 1,a).

Here we revisit this knowledge-intensive SFT data design that extends knowledge concepts to complete sequential text. The main advantages of this approach are: 1) Complete sequential text aligns with the fundamental learning objectives for LLMs as language models; 2) Natural language context provides information that helps models understand relationships between concepts, consistent with the spirit of instruction tuning (Ouyang et al., 2022); 3) The unified format facilitates alignment between test and training distributions.

However, large volumes of common text can shift the SFT dataset distribution away from the pre-training corpus, leading to the risk of overfitting to common styles and catastrophic forgetting(Liu et al., 2024; Li et al., 2024). This is also evidenced by the careful consideration for data variety (Han et al., 2024) and training techniques to mitigate catastrophic forgetting (Lu et al., 2025) when performing SFT.

Observe that a significant portion of common-style text derives from extended natural language contexts and model-specific formatting patterns. This observation prompts the question: Is it feasible

|  | Conventional SFT | Skip-Tuning |
|---|---|---|
| Training Data | $\langle|begin\_of\_text|\rangle\langle|start\_header\_id|\rangle$ user $\langle|end\_header\_id|\rangle$ 
 What is Cython influenced by ? $\langle|eot\_id|\rangle$ 
 $\langle|start\_header\_id|\rangle$ assistant $\langle|end\_header\_id|\rangle$ 
 Python $\langle|eot\_id|\rangle$ 
 **SFT Data (with Chat Template)** | Cython influenced by 

 Python 

 **Skip-Tuning Data** |
| Learning Correlations | user $\langle|end\_header\_id|\rangle \longrightarrow$ Cython influenced by 
 Cython influenced by $\longrightarrow$ Python 
 Python $\longrightarrow \langle|eot\_id|\rangle$ 
 $\cdot \cdot \cdot \cdot \cdot \cdot$ 
 **Correlations in Full Text** | Cython influenced by $\longrightarrow$ Python 

 **Focused Partial Correlation** |

Figure 1: Illustration of motivations for our Skip-Tuning work by highlighting the idea of partial-correlation learning. **a)** General SFT data formatted with a chat template. Black dashed blocks denote the model-specific formatting; blue dotted blocks denote the general natural language context; red blocks highlight the focused knowledge-intensive text segments. The demonstrated chat template uses LLaMA-3-8B-Instruct (AI@Meta, 2024). **b)** Correlations in the full SFT data, representing general co-occurrence patterns. **c)** Skip-Tuning data, which exclusively retains target knowledge-intensive texts. **d)** Correlations in the Skip-Tuning data, which represent the most important subset of the full correlations, thus forming the basis for what we call partial-correlation learning.

and advantageous to mitigate the potential biases introduced by these sources by using exclusively knowledge-intensive texts for SFT?

To be specific, our objective is not for the model to learn complete natural language expressions, which LLMs already excel, nor focus to learn model-specific formats, which typically occur frequently in fine-tuning datasets. Instead, our focus centers on knowledge-intensive content that may be underrepresented in standard training corpora, emphasizing the partial-correlations represented by specialized knowledge rather than numerous correlations represented by complete text (Illustrated in Figure 1).

To meet this intuition, in this paper, we introduce Skip-Tuning, a novel fine-tuning strategy for LLMs that utilizes non-successive text segments rather than conventional successive complete texts during the fine-tuning process. Skip-Tuning enables a paradigm that performs conventional language modeling within each text segment while modeling a skipped language modeling that captures the partial-correlation of the full text through the co-occurrence of text segments with certain position margins.

We will show that the partial-correlation learning intuition of Skip-Tuning is generally applicable, extending beyond knowledge injection settings. To validate the effectiveness of Skip-Tuning, we conduct experiments on both knowledge editing and system prompt following tasks, experimental results demonstrate improvements in fine-tuning effectiveness and generalization. Further, we show that Skip-Tuning enables a well-defined notion of position invariance, facilitating a design for robust learning.

Our contributions are summarized as follows,

- We propose Skip-Tuning, a novel fine-tuning strategy for LLMs that utilizes non-successive text segments. Skip-Tuning enables a partial-correlation learning paradigm for LLM fine-tuning, serves as an alternative or supplement to conventional SFT.

- We apply Skip-Tuning in the knowledge editing setting, by excluding common-style texts and using only knowledge-intensive text for fine-tuning, Skip-Tuning demonstrates improvements in fine-tuning effectiveness and generalization.

- We further demonstrate the effectiveness of partial-correlation learning in a system-prompt following task, which illustrates the broad application of Skip-Tuning across various NLP scenarios. Through this experimental setting, we discuss a general paradigm that enhance SFT performance with Skip-Tuning.

## 2 RELATED WORKS

**Supervised Fine-Tuning (SFT) for LLMs.**

SFT is the standard approach for adapting pretrained LLMs to specific tasks or domains. Early work by Ouyang et al. (2022) demonstrated that apply SFT with high-quality "instruction-response" pairs significantly improves model performance and generalization. Recent studies have explored various aspects of SFT, primarily focusing on data-based methods that apply domain-oriented data engineering, including data collection, filtering, and formatting to optimize model adaptation (Han et al., 2024; Chen et al., 2023). To achieve specific performance goals, synthetic data generation to improve SFT data quality has drawn significant attention (Xu et al., 2023; Sun et al., 2024), especially for specialized or complex tasks. These approaches are limited to complete sequential text, whereas our proposed Skip-Tuning method explores the potential of using non-successive text segments for fine-tuning.

**Knowledge Editing for LLMs.**

Knowledge editing for LLMs refers to the process of modifying or injecting specific knowledge (typically represented as knowledge graph triplets) into pre-trained LLMs. Locate-then-Edit is a representative category of methods that locate where the input knowledge is embodied in the model and edit the corresponding parameters(Meng et al., 2022a;b). Finetuning-based methods are considered as knowledge editing approaches for LLMs(Zhang et al., 2024a; Huang et al., 2025), particularly for parameter-efficient finetuning methods that enhance fine-tuning effectiveness while mitigating catastrophic forgetting(Hu et al., 2021; Lu et al., 2025). Unlike full retraining, these fine-tuning approaches modify only a small subset of parameters, thus aligning with the core principle of knowledge editing.

**System Prompt Following.**

Instruction following involves adhering to user intentions to generate helpful responses, which is fundamental to modern LLM applications (Zhang et al., 2024b). We focus on a specific case of instruction following that pays attention to the system prompt, which is a predefined directive guiding model behavior (Mukherjee et al., 2023). For chat models oriented toward customers, developers depend on system prompts to specify important context, output format, personalities, guardrails, content policies, and safety countermeasures (Mu et al., 2025).

## 3 PRELIMINARIES: INPUT FOR LLMS

Currently dominant structure of LLMs follows the decoder-only transformer architecture(Vaswani et al., 2017), which requires a tokenized text sequence as input. Such sequences consist of input IDs (X) containing tokens that encode character fragments, and position IDs (P) that encode token positions.

$$\mathbf{X} = [x_1, x_2, \ldots, x_L], x_i \in \mathbb{Z}$$

$$\mathbf{P} = [p_1, p_2, \ldots, p_L], p_i \in \mathbb{Z}$$

Our Skip-Tuning approach adopts the same input format but differs in the position IDs: whereas conventional SFT employs successive position IDs, Skip-Tuning utilizes non-successive position IDs.

## 4 METHOD: SKIP TUNING

In this section, we present Skip-Tuning, a fine-tuning strategy for LLMs that utilizes non-successive text segments rather than conventional successive complete texts. Skip-Tuning shares nearly the same modeling structure (Section 4.1) as conventional SFT on LLMs, except for data preparation and the processing of position IDs (Section 4.1.2). This approach enables a partial-correlation learning paradigm for LLM fine-tuning, we will discuss these applications in Section 4.2, including knowledge editing and system-prompt following tasks.

### 4.1 MODELING STRUCTURE

Skip-Tuning serves as a extension of LLMs SFT, which encourage the position IDs to be non-successive. As illustrated in Figure 2, the overall pipeline for Skip-Tuning includes data preparation, tokenization and position ID assignment, followed by the standard decoder-only transformer modeling(Vaswani et al., 2017).

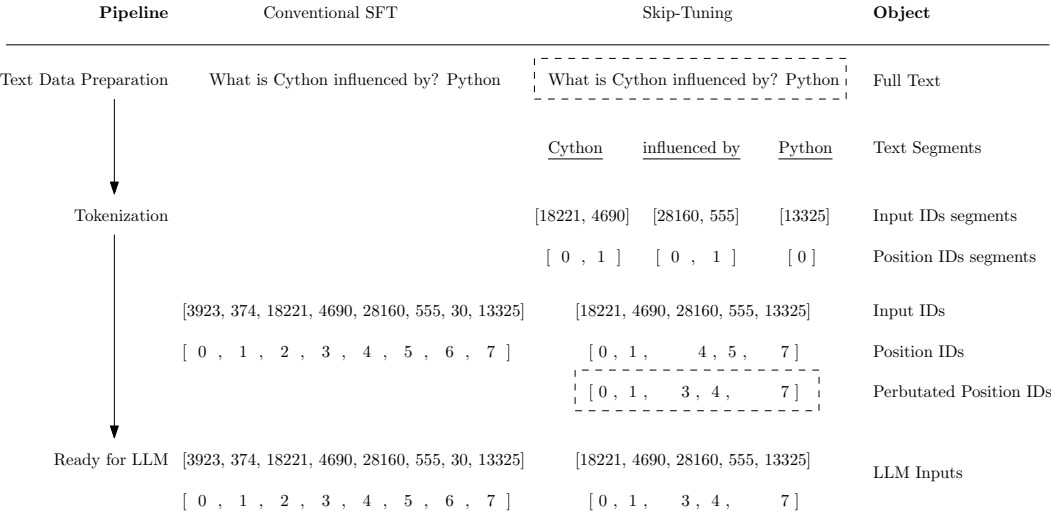

Figure 2: Modeling structure of Skip-Tunnig, compared with conventional SFT. Processes in dashed block are optional. The demonstrated tokenizatoin result is using LLaMA-3-8B-Instruct (AI@Meta, 2024) tokenizer.

### 4.1.1 DATA PREPARATION

Data preparation for Skip-Tuning involves collecting a list of ordered text segments, two major cases exist: 1)For tasks such as common knowledge editing setting with knowledge graph triplets, where training samples consist of non-successive text segments, it already fit the input for Skip-Tuning, further preparation is optional. As a comparision, conventional SFT generally requires synthesizing full sentences. 2) For tasks using successive text as training samples, Skip-Tuning involves splitting the text into segments based on the correlations between them that the model is required to learn.

### 4.1.2 POSITION IDS

To tokenize the text segments and form the input for LLMs, we first tokenize each text segment and then concatenate them to create the final input. To model the relative positions among segments, we assign position IDs to each segment in non-descending order. The non-successive nature of

Table 1: Training data illustration for transforming original data into Skip-Tuning data for knowledge editing and system prompt following tasks. Text segments are represented as comma-separated text lists. The chat template is formulated with message role in bracket "()"

| | ***For Knowledge Editing*** |
|---|---|
| Origin Data: | ["Cython", "influenced by", "Python"] |
| Skip-Tuning: | ["Cython", "influenced by", "Python"], ["influenced by", "Cython", "Python"] |
| | ***For System Prompt Following*** |
| Origin Data: | "(system)Think like you are answering to a five year old. |
| | (user)Write a sentence that describes: : John van den Brom, club, ADO Den Haag. |
| | (assistant)John van den Brom is a person who is associated with a club named ADO Den Haag." |
| Skip-Tuning: | ["(system)Think like you are answering to a five year old.", |
| | "John van den Brom is a person who is associated with a club named ADO Den Haag."] |

Skip-Tuning suggests using a predetermined margin for position IDs of each tokenized segment, the margin would be set according to the specific task requirements.

Here we discuss a subtle aspect specific to Skip-Tuning: position invariance. It is an intuition that slight disturbances in the input, including minor adjustments to word positions, would not be expected to significantly alter the output of LLMs. This is also a practical consideration in LLM prompt injection defense(Chen et al., 2024). In Skip-Tuning, this intuition is instantiated as position invariance, where given tokenized text segments with an arbitrary margin within a reasonable range, we expect the output to remain invariant, guaranteed by minor changes in the output representation.

This invariance could be categorized under geometric deep learning, where such properties are guarateed by modeling structure(Bronstein et al., 2021). In this work, we only involve a simple data-based intuitive method that randomly disturbs the predetermined skip margin, as reflected in the experiment on system prompt following (Section 5.2).

### 4.2 APPLICATIONS

In this section, we present two tasks for adopting Skip-Tuning to learn the expected partial-correlation. This covers application cases for both non-successive knowledge-intensive text (knowledge editing) and complete sequential text (system prompt following). The design of Skip-Tuning samples for these tasks is illustrated in Table 1.

#### 4.2.1 KNOWLEDGE EDITING

For the common knowledge editing setting involving a knowledge graph triplet $(subject, relation, object)$ (Huang et al., 2025), the triplet itself naturally serves as the target text segments for Skip-Tuning, representing a declarative sentence structure. We employ both the original triplet $(subject, relation, object)$ and its swapped counterpart $(relation, subject, object)$ as Skip-Tuning samples, based on the intuition that the variation in word order between interrogative and declarative sentence structures. We suggest a moderate position ID margin, following the intuition that these knowledge segments generally co-occur closely within the text.

#### 4.2.2 SYSTEM PROMPT FOLLOWING

For system prompt following tasks, we follow the general LLM Supervised Fine-Tuning (SFT) setting, which utilizes high-quality instruction tuning datasets formatted with chat templates for fine-tuning. We randomly substitute a portion of the training samples with their corresponding Skip-Tuning samples.

Here we discuss the design of these Skip-Tuning samples. The SFT data for system prompt following generally involves a system prompt message, followed by a list of user-model interaction. This data is generally formatted as the widely adopted "system-user-assistant" message list for commercial and open-source LLMs (OpenAI, 2022; AI@Meta, 2024). The goal of this setting is to develop instruction-following capabilities for LLMs, with special emphasis on following the system prompt.

To strengthen the correlation between system instructions and model responses, we employ Skip-Tuning that exclusively selects system prompts and model responses from the training data, while

omitting the user prompts. To reserve the positional slots of the omitted user prompt, a larger position ID margin is suggested.

# 5 EXPERIMENTS

## 5.1 KNOWLEDGE EDITING

### 5.1.1 DATASETS AND TASKS

To demonstrate the effectiveness of adopting Skip-Tuning for knowledge injection, we conduct experiments on the knowledge editing benchmark HalluEditBench(Huang et al., 2025). HalluEdit-Bench assesses editing performance across multiple dimensions, including Efficacy (direct edit), Generalization (edit generalization across different form), Portability (multi-hop edit efficacy), and Locality (degree of forgetting). This benchmark specifically addresses hallucinations in LLM generations by evaluating knowledge injection methods designed to reduce such errors.

HalluEditBench provides training data in two forms: a knowledge graph triple and a corresponding sentence that represents it. These formats are suitable for Skip-Tuning and conventional SFT, respectively, allowing for a comparison between these approaches.

### 5.1.2 EXPERIMENTAL CONFIGURATION

We use LLaMA-3-8B-Instruct(AI@Meta, 2024), LLaMA-2-7B-chat(Touvron et al., 2023) and Mistral-v0.3-Instruct-7B(Jiang et al., 2023) as base models, and fine-tune for 64 steps each with the parameter-efficient fine-tuning method LoRA(Hu et al., 2021), with rank $r = 8$, targeting for Q,V matrix in the attention layer the transformer. The learning rate is set to 5e-4 for LLaMA-3-8B-Instruct and 2e-4 for LLaMA-2-7B-chat and Mistral-v0.3-Instruct-7B. The predetermined margin for Skip-Tuning is set to 5. Experiments are conducted on 1 NVIDIA A100 40G GPU.

### 5.1.3 COMPARISON METHODS

We aim to compare the performance of SFT and Skip-Tuning (denoted as "-skip") for knowledge injection. Conventional SFT for knowledge editing pose a risk of catastrophic forgetting, where base model capabilities may lost during the adaptation process. Besides the LoRA fine-tuning, we include results of CLoRA(Lu et al., 2025), which is a learning-based catastrophic forgetting mitigation method that introduces no impact on data utilization, making it suitable for our comparative analysis.

We include a ablation result for using $(subject, relation, object)$ for Skip-Tuning data alone, denoted as "-noswap", instead of both $(subject, relation, object)$ and $(relation, subject, object)$ for the default setting.

### 5.1.4 RESULTS AND ANALYSIS

**Overall Results**  We report the results on HalluEditBench in Table 2. Overall, Skip-Tuning demonstrates superior or comparable results for all assessment dimensions. The result demonstrates the feasibility of Skip-Tuning, especially considering that we are using lossy data with reduced formatting quality.

**Effectiveness Aware Metrics**  Efficacy and Generalization are two direct metrics that reflect the fine-tuning effectiveness. Results show that Skip-Tuning generally achieves improvement across these metrics. Note that for our included CLoRA result, Generalization shows improvement at the cost of lower Efficacy, we owe this to the constraints on learning. In contrast, when introducing Skip-Tuning, Efficacy and Generalization simultaneously improve, which demonstrates the effectiveness of Skip-Tuning as a knowledge injection strategy that successfully balances efficacy and generalization.

**Forgetting Aware Metrics**  Localization is the most direct metric for measuring forgetting, as it quantifies the model's ability to recall information beyond the edit target. Meanwhile, Multi-Hop

Table 2: Evaluation results on HalluEditBench benchmarks, with accuracy scores (%) reported for the four assess dimension: Efficacy, Generalization, Multi-Hop Portability, and Locality. **Bold** font highlight the best result among all compared results for same base model and fine-tuning method.

| Methods | Efficacy | Generalization | Multi-Hop | Locality |
|---|---|---|---|---|
| *LLaMA-3-8B-Instruct* | | | | |
| LoRA | 98.09 | 49.93 | 19.09 | 14.57 |
| LoRA-skip | **98.53** | 52.68 | **23.17** | **22.40** |
| LoRA-skip-noswap | 98.49 | **52.73** | 22.12 | 22.31 |
| CLoRA | 90.53 | **53.43** | **25.81** | **40.75** |
| CLoRA-skip | **94.95** | 52.42 | 23.39 | 35.15 |
| CLoRA-skip-noswap | 94.46 | 52.42 | 23.39 | 40.68 |
| *LLaMA-2-7B-Chat* | | | | |
| LoRA | 91.85 | 45.74 | **24.44** | **48.51** |
| LoRA-skip | 96.76 | **47.99** | 17.98 | 38.89 |
| LoRA-skip-noswap | **97.10** | 48.08 | 17.98 | 39.44 |
| *Mistral-v0.3-7B-Instruct* | | | | |
| LoRA | **98.68** | 48.57 | 13.42 | **25.16** |
| LoRA-skip | 96.14 | **52.06** | 14.67 | 22.29 |
| LoRA-skip-noswap | 95.60 | 48.57 | **16.55** | 24.84 |

metric involves inference capabilities that are more closely related to the base model's inherent reasoning abilities. These two metrics provide an evaluation for the forgetting.

Results show that Skip-Tuning generally out-performs on LLaMA-3-8B-Instruct and Mistral-v0.3-7B-Instruct, but under-performs on the weaker base model LLaMA-2-7B-chat and when compared to CLoRA results. It is worth noting that these under-performed results occur in situations where Efficacy largely under-performs Skip-Tuning, as the weaker base model or the added learning constraints by CLoRA resist learning. Thus, we attribute this to the fact that with strong learning intensity, Skip-Tuning helps improve robustness and mitigate forgetting.

**Whether to Swap Subject and Relation?** With both $(subject, relation, object)$ and $(relation, subject, object)$, the results showed generally improved efficacy, though not significantly, which reflects our intuition that the order with relation before subject is more common. herefore, we generally recommend adopting both order when using Skip-Tuning to learn knowledge graph triplets.

## 5.2 SYSTEM PROMPT FOLLOWING

### 5.2.1 DATASETS AND TASKS

To demonstrate the effectiveness of adopting Skip-Tuning for system prompt following, we conduct experiments on two benchmarks: RealGuardRails and System-IFEval.

RealGuardRails, a recently proposed system prompt following benchmark(Mu et al., 2025), consists of 239 handwritten test cases that either align or conflict with the system prompt in each test case, and 504 distractor test cases which attempt to distract the model away from its system prompt with in-context demonstrations of unrelated tasks. We use official suggested GPT-4o-2024-08-06(OpenAI, 2024) as the required LLM-as-judge model.

System-IFEval, suggested by Mu et al. (2025), is based on the instruction-following benchmark IFEval(Zhou et al., 2023b), and transform the rules in user input to system prompt, thus introduces an emphasis on system prompts. We use System-IFEval to evaluate the ability of LLMs to follow precise instructions embedded in their system message.

We use the training dataset from Mu et al. (2025) for fine-tuning, which is a high-quality SFT dataset composed of 151K data samples covering single-turn and multi-turn, generic and complex system prompts. We transform part of the data sample with system prompt to Skip-Tuning sample. More details for the training data configuration are listed in Appendix A.

Table 3: Evaluation results on System-IFEval and RealGuardRails benchmarks, with accuracy scores (%) reported. For System-IFEval, "[S]" and "[L]" denote strict and loose accuracy, "P" and "I" indicate the prompt and instruction level. Underlined text highlight best result for the same Skip-Tuning ratio. **Bold** font highlight the best result among all compared results.

| Methods | System-IFEval | | | | RealGuardRails | |
|---|---|---|---|---|---|---|
| | P[S] | I[S] | P[L] | I[L] | handwritten | distractors |
| SFT(Mu et al., 2025) | 59.40 | - | 62.30 | - | 46.00 | 24.60 |
| skip-0% (SFT) | 60.64 | 71.27 | 64.89 | 74.34 | 44.35 | 39.48 |
| skip-25% | | | | | | |
|   disturb-50% | 64.04 | **73.78** | **68.51** | **76.85** | **51.46** | 44.84 |
|   disturb-25% | 61.28 | 71.55 | 65.11 | 74.48 | 47.70 | 44.64 |
|   disturb-0% | 61.49 | 70.57 | 65.32 | 73.78 | 46.03 | 43.25 |
| skip-50% | | | | | | |
|   disturb-50% | 62.34 | 71.97 | 67.02 | 75.45 | 51.05 | 41.87 |
|   disturb-25% | 63.19 | 72.26 | 67.02 | 75.45 | 48.12 | 41.67 |
|   disturb-0% | **64.26** | 73.08 | 67.66 | 76.01 | 44.77 | 42.06 |
| skip-75% | | | | | | |
|   disturb-50% | 58.94 | 68.76 | 64.04 | 73.08 | 42.68 | 38.69 |
|   disturb-25% | 59.36 | 68.76 | 64.47 | 72.66 | 45.61 | 40.28 |
|   disturb-0% | 60.43 | 70.43 | 65.11 | 74.20 | 44.77 | 38.10 |

### 5.2.2 EXPERIMENTAL CONFIGURATION

We use LLaMA-3-8B(AI@Meta, 2024) as the base model and fine-tune the dataset for 2 epochs with the parameter-efficient fine-tuning method LoRA(Hu et al., 2021), with rank $r = 16$, targeting all attention and FFN layers in the transformer. The learning rate is set to 1e-4 with a cosine scheduler, the max training length is set to 4096 and over-lengthed data are dropped, the batch size is set to 128. The predetermined margin for Skip-Tuning is set to 100. Experiments are conducted on 2 NVIDIA A100 40G GPUs.

### 5.2.3 COMPARISON METHODS

We aim to test the effectiveness of introducing partial-correlation learning through Skip-Tuning, compared to conventional SFT. As mentioned in the method section 4.2.2, we randomly substitute a portion of the training samples with their corresponding Skip-Tuning samples. The retained SFT samples ensure the learning for common generation capability.

To evaluate the impact of Skip-Tuning, we control the ratio of Skip-Tuning samples to test the benefits and trade-offs of introducing Skip-Tuning samples in the system prompt following SFT. We denote the variant **skip-x%** as substituting x% of substitutable samples with Skip-Tuning samples. Additionally, we examine Position ID disturbance as a data augmentation mechanism to improve generalization. We test its effectiveness with varying disturbance ranges, denoting the variant **disturb-x%** as randomly disturbing position IDs with margin $m$ in the range of $((1 - x\%)m, (1 + x\%)m)$. We include result in (Mu et al., 2025), which is full parameter fine-tuned, our re-implement reaches a higher result.

### 5.2.4 RESULTS AND ANALYSIS

We report the results for System-IFEval and RealGuardRails benchmarks in Table 3, which presents the main results for using different Skip-Tuning sample ratios, along with ablation results for position ID disturbance.

**Impact of Skip-Tuning Ratio** The Skip-Tuning setting we proposed for the system prompt following task may seem like an uncommon design, as it breaks the successive generation of model output in response to user input. This characteristic raises questions about the compatibility of Skip-Tuning with conventional SFT, necessitating an investigation into mixed training approaches. Our results about these concerns with two key findings: 1) Mixed training of Skip-Tuning and conven-

tional SFT is both feasible and advantageous; 2) A certain portion of conventional SFT data needs to be retained to maintain optimal performance.

For 1), the superior results for Skip-Tuning with 25% and 50% ratios compared to SFT demonstrate the effectiveness of Skip-Tuning for the system prompt following task. Even with a large ratio of 75%, the capability for generation in response to system and user prompts is retained, showing a performance comparable to SFT. This demonstrates the compatibility of conventional SFT and Skip-Tuning, highlighting that Skip-Tuning serves as a feasible and advantageous supplement to conventional SFT.

For 2), although performance comparable to SFT with a large ratio 75% of Skip-Tuning substitution, it remains below the SFT baseline. Also, best result is get on 25%, demonstrate This highlights the need for a certain portion of SFT data to learn the full relationships within the text that maintain generation quality.

**Impact of Position ID Disturbance**   The position ID disturbance is introduced for Skip-Tuning as an intuitional utilization for position invariance4.1.2. Practically, it serves as a data-augmentation strategy for Skip-Tuning to facilitate robust learning and generalization. As the results show, for skip-25%, the best performance is achieved with large disturbance, while for larger skip ratios, the advantage is diminished. We attribute this to the fact that with large Skip-Tuning ratios, there is sufficient variety in the Skip-Tuning samples. Consequently, the disturbance introduces fewer advantages and is overshadowed by the disadvantages of insufficiently learning these samples. For small skip ratios, however, the position ID disturbance is recommended.

### 5.2.5 Discussion

The proposed and experimentally tested system prompt following tasks demonstrates a potential general paradigm for applying Skip-Tuning to enhance SFT performance, consists of steps as follows:

1. Identify the core capability that the existing SFT data is expected to enhance, such as system prompt following capability here.
2. Select text segments that most essentially demonstrate this capability through their correlation, and predetermine the margin between each two successive segments to reflect position-aware co-occurrence.
3. Train with a mixture of original SFT samples and Skip-Tuning samples, applying position ID perturbation if needed.

## 6 Conclusion and Limitations

In this paper, we introduce Skip-Tuning, a novel fine-tuning strategy for LLMs that uses non-successive text segments instead of complete sequential texts. Skip-Tuning performs skipped language modeling on text segments, enabling a paradigm of partial-correlation learning where the model learns from sparse but meaningful text fragments. Experimental results demonstrate the effectiveness of Skip-Tuning on knowledge editing and system-prompt following tasks. Skip-Tuning shows potential for broader application across various NLP scenarios. We introduce a "Target Capacity - Reflected Text Fragment - Mixture Skip-Tuning with SFT" pipeline that acts as a general paradigm for enhancing SFT performance with Skip-Tuning.

Limitations of this paper include: 1) Broader Applications: Our empirical tests for Skip-Tuning cover only one scenario for both with non-successive and successive training data. Exploring more applications would be valuable, and we leave this to future work. 2) Deeper Insight with Position Invariance: The position invariance for Skip-Tuning discussed in this paper may play an important role in improving the robustness of LLMs, but we only provide a data-based method that intuitively meets this requirement. Architecture-based methods for achieving position invariance may be a future direction. The value of this approach would be particularly highlighted in custom-oriented scenarios, which we leave for future work.

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

## A  Detailed Experiment Setup for System Prompt Following

### A.1  Training Dataset

We use the official training dataset from the RealGuardRails benchmark (Mu et al., 2025). This dataset is a high-quality SFT dataset comprising 151K data samples that cover both single-turn and multi-turn conversations, as well as generic and complex system prompts. The distribution of these data types is presented in Table 4.

Table 4: Training Data in RealGuardRails benchmark.

| Data | Quantity | Description |
|---|---|---|
| RealGuardrails SFT | 18497 | single-turn, tool-calling assistants, system prompts |
| Multifaceted Collection | 20000 | single-turn, complex persona system prompts |
| Glaive v2 | 20000 | single-turn, tool-calling, system prompts |
| SPML | 12541 | single-turn, system prompts, prompt injection attempts with newly-generated completions |
| Tulu3 Persona IF | 20000 | single-turn, instruction-following |
| Tulu3 WildGuardMix | 20000 | single-turn, harmful/benign refusals and responses |
| WildChat GPT-4 | 20000 | multi-turn, real user conversations with GPT-4 |
| SlimOrca | 20000 | single-turn, instruction + CoT answer, generic system prompts |

Out of 151K data samples, 91K contain system prompts. In our Skip-Tuning data processing design (Section 4.2.2), only data containing system prompts can be transformed. Therefore, substituting 50% of the substitutable data accounts for nearly 30% of the full dataset size.

## B  Reproducibility Statement

We have discussed the hyperparameter settings in the experiment section 5. The proposed Skip-Tuning method is easy to implement and requires almost no modification to be adopted in various LLM infrastructure settings. The evaluation results for benchmarks were obtained using the official code.

## C  The Use of Large Language Models (LLMs)

We utilized the GLM-4.5(ZhipuAI, 2025), a commercial large language model (LLM), to polish the writing. Our usage involved these stages: initially, we manually drafted the paper's paragraphs, and subsequently we employed the LLM to rewrite them for spelling and grammar correction. We meticulously reviewed the generated output to ensure it remained consistent with our original content and intended meaning.

