# OpenReview forum: "Partial-Correlation Learning for Large Language Models with Skip-Tuning"
_ICLR.cc/2026/Conference — ICLR 2026 Conference Withdrawn Submission_

### Official Review · Reviewer_5EJp · 2025-10-29

**Soundness:** 2
**Presentation:** 3
**Contribution:** 2
**Rating:** 4
**Confidence:** 4

**Summary:**

This paper proposes Skip-Tuning, a novel fine-tuning strategy for Large Language Models (LLMs) that utilizes non-successive text segments instead of complete sequential text. The method enables a paradigm called partial-correlation learning, allowing models to focus on sparse but knowledge-rich fragments while skipping redundant or stylistically common text. The authors validate Skip-Tuning on knowledge editing and system prompt following tasks.

**Strengths:**

1. Innovative Concept – The idea of learning from "partial correlations" through skipped text segments introduces a fresh perspective on data-efficient fine-tuning for LLMs.
2. Implementation Simplicity – Skip-Tuning requires only minor changes (adjusting position IDs), making it lightweight and practical to deploy.
3. The paper is clearly written and easy to follow.

**Weaknesses:**

**Limited Empirical Strength**. The results in Table 2 primarily demonstrate the feasibility rather than the superiority of Skip-Tuning. For CLoRA-based LLaMA-3-8B-Instruct, LoRA-based LLaMA-2-7B-Chat, and Mistral-v0.3-7B-Instruct, Skip-Tuning does not yield consistent improvements across overall metrics (e.g., higher edit effectiveness often comes with greater forgetting). This indicates that while the method works, it is not convincingly better than standard SFT or LoRA. Consequently, the claim of "improvement in fine-tuning effectiveness and generalization" appears somewhat overstated given the presented performance.

I value the conceptual novelty of this approach and its promising results in system prompt following. However, its performance on knowledge editing is less convincing. Knowledge editing is one of the most suitable application for Skip-Tuning, since knowledge injection does not strongly depend on input format. The results in this setting suggest that the practical benefit of the method may be limited.

**Questions:**

None

---

### Official Review · Reviewer_SGN4 · 2025-10-30

**Soundness:** 2
**Presentation:** 2
**Contribution:** 2
**Rating:** 4
**Confidence:** 4

**Summary:**

The paper proposes Skip-Tuning, a fine-tuning strategy for LLMs that replaces conventional sequential text inputs with non-successive text segments. The core idea is to perform partial-correlation learning from sparse but meaningful knowledge-intensive fragments, thereby alleviating distribution shift from pretraining corpora and reducing catastrophic forgetting. The effectiveness is evaluated on knowledge editing and system-prompt following tasks, showing comparable performance across multiple models and settings.

**Strengths:**

1. The proposed Skip-Tuning requires almost no modiﬁcation to be adopted in various LLM infrastructure settings.

2. Experiments across knowledge editing and system-prompt tasks demonstrate generality.

**Weaknesses:**

1. The methodology about selecting text segments, determining positional margins, and assigning position identifiers requires more detailed explanation.

2. Some sentences are hard to follow. The description of the method, particularly in Sections 4.1.2 and 4.2, could be more structured for easier understanding.

3. The impact of the predetermined margin between segments is not explored in depth. The choice of margins (5 for knowledge editing, 100 for prompt following) seems somewhat arbitrary.

4. A deep theoretical explanation or analysis of why and how this method works from partial-correlation learning is missing.

**Questions:**

1. How sensitive are the results to the choice of the position ID "margin"? Are there guidelines or principles  for setting this margin?

2. How would the method perform if the segments were chosen arbitrarily?

3. If the tasks are without a predefined structure, how is the margin between segments determined?

4. Under which scenarios or types of tasks do you anticipate Skip-Tuning might perform poorly?

**Details Of Ethics Concerns:**

N.A.

---

### Official Review · Reviewer_j1GE · 2025-10-31

**Soundness:** 2
**Presentation:** 2
**Contribution:** 2
**Rating:** 2
**Confidence:** 3

**Summary:**

This paper introduces Skip-Tuning, a fine-tuning strategy for large language models that uses non-successive text segments rather than full sequential text for supervised fine-tuning. The authors argue that conventional SFT datasets often contain large amounts of “common-style” or formatting-heavy text that may cause distribution shift and increase the risk of catastrophic forgetting. Skip-Tuning instead focuses on knowledge-intensive fragments and assigns non-successive positional IDs to encourage the model to learn partial-correlation patterns rather than full contextual co-occurrences.

The approach is evaluated in two settings: knowledge editing (using HalluEditBench) and system prompt following (using RealGuardRails and System-IFEval). Across several base models (LLaMA-2, LLaMA-3, Mistral), Skip-Tuning shows improvements in efficacy and generalization, with some evidence of reduced forgetting in stronger base models. In system prompt following, mixing Skip-Tuning samples with standard SFT improves adherence to system instructions, especially when using moderate Skip-Tuning ratios (e.g., 25–50%) and optional position ID perturbation.

Overall, the paper shows some empirical gain without having any analytical insight into why it's the case.

**Strengths:**

The paper introduces a simple and practical method. Skip-Tuning requires minimal architectural changes and integrates naturally with existing LLM fine-tuning workflows. It is easy to adopt in practice.

The papers demonstrates broad applicability. The method is tested on both knowledge editing and system prompt adherence, which strengthens the claim that Skip-Tuning is not task-specific.

The paper shows gains in some key empirical metrics. Improvements in efficacy and generalization (in HalluEditBench) and system prompt compliance (in RealGuardRails/System-IFEval) suggest that partial-correlation learning is meaningful.

**Weaknesses:**

The paper lacks theoretical grounding. While the intuition of “partial-correlation learning” is appealing, the paper does not provide a formal characterization of what correlations are preserved vs. removed, or how this affects the model’s internal representation. The explanation remains qualitative.

The paper shows mixed forgetting results. While Skip-Tuning improves forgetting metrics for strong models, it underperforms on weaker models and when interacting with CLoRA constraints. This suggests that the method’s robustness is not consistent across architectures.

Evaluation on System Prompt Following Could Be Expanded. The benchmarks used measure rule adherence, but do not evaluate fluency, safety, or stability trade-offs introduced by fragment-based tuning.

**Questions:**

None

---

### Official Review · Reviewer_eyaG · 2025-11-01

**Soundness:** 3
**Presentation:** 2
**Contribution:** 2
**Rating:** 4
**Confidence:** 3

**Summary:**

This paper proposes an input-enhanced training approach that improves the generalization of SFT by fine-tuning LLMs on selected meaningful text fragments from the training data. The experimental results demonstrate improvements in both knowledge editing and system prompt following.

**Strengths:**

* The proposed method is relatively novel.
* The experiments demonstrate the superiority of the proposed approach on knowledge editing and system prompt following.

**Weaknesses:**

* The proposed method bears similarity to data augmentation techniques for LLMs. However, the paper lacks sufficient comparisons with related baseline methods (e.g., traditional robust training approaches that introduce perturbations in the input space such as synonym substitution), and the corresponding discussion is limited. The absence of such comparison and discussion is my main concern about this paper.

* For instruction-following tasks, the proposed method requires tuning two important hyperparameters, yet the paper only investigates its performance on a single model, which limits the generalizability of the method.

**Questions:**

* Can the proposed method be combined with ICL? Extending it to a broader range of SFT tasks would be valuable.

* How would the performance change if traditional perturbation techniques, such as synonym substitution or random token deletion on non-meaningful text fragments, were applied instead?

---

### Note · Authors · 2025-12-05

I have read and agree with the venue's withdrawal policy on behalf of myself and my co-authors.